# Quantification of Pork, Chicken, Beef, and Sheep Contents in Meat Products Using Duplex Real-Time PCR

**DOI:** 10.3390/foods12152971

**Published:** 2023-08-07

**Authors:** Yanwen Wang, Emily Teo, Kung Ju Lin, Yuansheng Wu, Joanne Sheot Harn Chan, Li Kiang Tan

**Affiliations:** 1National Centre for Food Science, Singapore Food Agency, 7 International Business Park, Singapore 609919, Singapore; wang_yanwen@sfa.gov.sg (Y.W.);; 2Department of Food Science and Technology, National University of Singapore, S14 Level 5 Science Drive 2, Singapore 117542, Singapore

**Keywords:** meat safety, food fraud, meat speciation, meat quantification, duplex qPCR, chicken, pork, beef, sheep

## Abstract

Accurate methods for meat speciation and quantification are essential for ensuring the supply of safe and wholesome meat and composite products with animal origins to negate the potential associated hazards, aid classification of consignments at the import control system, and thwart food fraud committed for financial gain. To better enhance meat safety control and combat food fraud, this study developed two duplex real-time polymerase chain reaction (real-time PCR) systems specifically designed for chicken, pork, sheep, and beef, using single-copy, chromosomally encoded, species-specific gene sequences to accurately measure the content of each meat type in meat products. DNA extracted from the raw and boiled reference materials prepared in varying proportions (ranging from 1% to 75%) were used in the development of the duplex assay to derive calibration factors to determine the meat content in different meat products. The method was further validated using proficiency test samples and market monitoring samples. Our findings showed that this method exhibits high specificity and sensitivity, with a significant accuracy range of 0.14% to 24.07% in quantifying the four meat types in both raw and processed meat products. Validation results further confirmed the effectiveness of our method in accurately quantifying meat content. Thus, we have demonstrated the duplex qPCR assays as promising approaches for implementation in routine analysis to strengthen meat safety control systems and combat meat fraud, thereby safeguarding consumer health and trust in the meat industry.

## 1. Introduction

Globally, the growing demand for meat consumption and changing trends in production and supply in the past five decades pose as new challenges in food safety and public health concerns [1,2]. Meat of animal origins can potentially harbour a plethora of pathogens including bacteria, virus, and parasites leading to zoonotic diseases [2,3,4,5]. It is estimated that approximately millions of deaths and one billion cases of illness are in connection with zoonoses annually [4]. Therefore, the establishment of a stringent system involving inspection, quarantine, and testing of meat, composite meat products, and other products of animal origins is of pivotal importance for effective crossborder control and management.

Meat speciation and quantification testing serves as an early intervention at import checkpoints to ensure conformities of composite products containing processed products of animal source as well as identification of noncompliant compound meat products. Competent food safety authorities would lay down specific threshold values to impart clarity in the product classification in the custom system. Stipulated criteria with appropriate prescribed inspections and quarantine requirements, together with testing results as indicative evidence, facilitate effective veterinary checks and controls at crossborders [6,7,8,9]. For example, according to European Commission Regulation 2021/632, any composite food products containing over 20% by weight of sausage, meat, meat offal, and other ingredients of animal origins must be subjected to official controls at border [6]. In the United States, the Food Safety and Inspection Service (FSIS) has established guidance values that allows the exemption of food products containing no more than 3% raw meat, or no more than 2% cooked meat or carcass, or containing no more than 2% cooked poultry meat [8]. Correspondingly, under Singapore’s General Classification of Food & Food Products, food products containing more than 5% meat contents would be classified as meat products [9], for which importers need to carry a valid license to import such products and the related consignments would be subject to import control requirements for meat products under the regulations of the Wholesome Meat and Fish Act [10,11].

Besides supporting crossborder checks, meat speciation and quantification testing are also commonly conducted in other contexts to enable the investigation of food fraud committed for economic gain, the examination of labelling compliance for Halal food or vegetarian food, or purely for product quality checks [12,13,14]. Therefore, reliable testing of meat speciation and meat content is a critical laboratory capability for ensuring food safety, public health, as well as for supporting food fraud investigations and quality control.

Currently, methods for identifying animal species in food products are primarily based on deoxyribonucleic acid (DNA) or protein analysis [12]. Protein-based methods have limitations when used for the testing of highly processed foods due to the high heat or pressure involved in meat production [13,14]. Moreover, protein-based methods are not suitable for distinguishing closely related animal species [12,13]. Molecular-based techniques, such as real-time polymerase chain reaction (PCR), digital PCR, and Next-Generation-Sequencing (NGS), are widely utilized for identifying and quantifying animal species in food products attributing to the relative thermal stability of DNA [13,14,15,16]. Particularly, TaqMan real-time PCR assays have revolutionized nucleic acid analysis, offering improved accuracy, sensitivity, and efficiency in identifying and quantifying specific meat species in food products compared to traditional PCR methods. Numerous studies utilizing TaqMan real-time PCR assays, including singleplex reactions to multiplex systems, have been used for different food products’ quantification [13,17,18,19]. However, the accurate measurement of meat content in food samples, especially for enforcement purposes, remains challenging due to the complex tissue types used in food production [20].

To overcome the challenge of accurate quantification of meat proportions in food samples, one common approach is to establish a correlation between DNA content and actual meat proportions for each meat species, known as calibration factors [20,21,22,23,24,25]. By using these calibration factors specific to each meat species, the DNA content can be converted into meat content, eliminating the needs of reference material preparation in subsequent analyses [19,22,23,24,25]. Several studies have successfully estimated the proportions of meat content in food samples using multiplex quantitative PCR (qPCR) combined with calibration factors [19,23,25]. However, the precision of quantifying meat proportions using these multiplex qPCR systems is limited, exhibiting a range of accuracy from 1% to 167% [19,23,25]. Such an extensive range of accuracy is insufficient for the regulatory authorities and enforcement bodies responsible for meat import control to have an effective oversight and follow-up actions. Furthermore, calibration factors derived from different laboratory could vary which warrants every laboratory to determine its own calibration factors for each animal species and sample processing workflow [20,25].

Therefore, to address the need for a highly accurate method that meets the stringent requirements of regulatory authorities overseeing meat safety control, this study aimed to develop two duplex qPCR systems to accurately measure the proportion of meat fractions in processed meat products. The duplex assays were specifically developed for the four primary sources of meat for local consumption, namely pork, beef, chicken, and sheep [26]. The proposed method involved the use of raw and boiled reference materials with known proportions, enabling the derivation of calibration factors for each meat species. The method was further validated using proficiency test samples and market meat samples to demonstrate the feasibility and potential of this method for routine meat safety control analysis.

## 2. Materials and Methods

### 2.1. Sample Preparation

Frozen lean meat samples of pork (*Sus scrofa*), sheep (*Ovis aries*), beef (*Bos taurus*), and chicken (*Gallus gallus*) were collected from retail outlets and the authenticity of these samples was verified using the RapidFinder™ Meat ID kits for pork, chicken, beef, and sheep (Thermo Fisher Scientific Inc., Waltham, MA, USA). Reference mixtures containing 1%, 9%, 15%, 25%, 50%, and 75% (*w*/*w*) of pork, beef, chicken, and sheep were prepared according to Table 1. To simulate industrially processed meat products, a second set of mixtures (with the same concentrations in Table 1) was prepared and autoclaved at 121 °C for 15 min. A total of 100 g raw and autoclaved meat samples for each species were separately homogenized using a blender (Grindomix GM 200; Retsch, Haan, Germany) and dried in an oven (Memmert, Schwabach, Germany) at 60 °C for 72 h. The dried samples were homogenized again using the blender and then minced to a superfine powder in liquid nitrogen by using a pestle and mortar. Subsequently, the mixtures specified in Table 1 were prepared, with each mixture reaching a final weight of 200 mg. To ensure that the extracted DNA accurately represented the proportions of different meats, the meat mixtures were further ground using a TissueLyser II (Qiagen, Hilden, Germany) at 30 times per second for 1 min until mixed evenly.

A total of twelve retail meat samples and three proficiency test (PT) samples from the Food Analysis Performance Assessment Scheme (FAPAS) are used for method validation (Appendix A). The samples were homogenized similarly to the reference mixtures. The blender and other accessories were cleaned, immersed in a bleaching agent (sodium hypochlorite 20%) for 30 min and were washed to remove residual DNA before use for homogenization. To prevent the enzymatic degradation of DNA, both prepared samples and extracted DNA were immediately stored at −20 °C and portioned to avoid repeated freeze–thawing cycles.

### 2.2. DNA Extraction

The DNA from meat samples was extracted using the DNeasy Blood & Tissue Kit (Qiagen, Hilden, Germany) according to the manufacturer’s instructions with minor adjustments. Specifically, 200 mg of ground sample material was extracted, and the DNA was eluted into 100 μL elution buffer according to the supplier’s manual. The extraction of plant DNA used for specificity tests was carried out using the cetyltrimethylammonium bromide (CTAB) protocol described in EN ISO 21571 [27]. The concentration and purity of isolated DNA were determined by measuring the absorbance at 260 nm (A260) and 280 nm (A280) using a NanoDrop™ 2000 Spectrophotometer (Thermo Fisher Scientific, Waltham, MA, USA). The extracted DNA was aliquoted and stored at −20 °C.

### 2.3. Primers and Probes

The primers and probes utilized in this study have been previously reported and their details are presented in Table 2. To determine the amplifiability of DNA extracts and eliminate any false-negative or false-positive results of the duplex qPCR assays detection, a eukaryotic-reference system using the 18S ribosomal RNA (18S rRNA) gene fragment found in all eukaryotic cells was considered as an endogenous control. The primer and probe sets employed in this study all target single-copy, chromosomally encoded gene sequences. These primer and probe sets were synthesized by GENEWIZ (Azenta Life Sciences, Suzhou, China).

### 2.4. Real-Time PCR Procedure

The real-time PCR was carried out using the Applied Biosystems™ QuantStudio™ 6 Pro Real-Time PCR System (Thermo Fisher Scientific Waltham, MA, USA) in an optical 96-well reaction plate (0.2 mL) sealed with optical adhesive film (Thermo Fisher Scientific, Waltham, MA, USA). The PCR was performed in a total volume of 25 μL, with the reaction mixture consisting of 12.5 μL of TaqMan™ Environmental Master Mix 2.0 (Thermo Fisher Scientific, Waltham, MA, USA), 4.75 μL of nuclease-free water, 2.5 μL of 10x primer/probe mix (Table 2), 0.25 μL of Uracil-DNA glycosylase (UNG) (New England Biolabs, Ipswich, MA, USA), and 5 μL of isolated DNA. The amplification program included a 2 min hold at 50 °C, an initial denaturation step at 95 °C for 10 min, followed by 45 cycles of amplification at 95 °C for 15 s and 60 °C for 1 min.

### 2.5. Amplification Efficiency and Quantification

To determine the amplification efficiency, the standard curves which are considered as an indicator for real-time PCR performance evaluation were plotted using the logarithm of the studied concentrations in the dilution series (Table 3) against the mean Ct values of the assays. The linear regression equation of the models is described as follows: y = ax + b, where the x-axis is the log DNA amount for each meat species, the y-axis represents the mean Ct values, and “a” and “b” are the slope and intercept, respectively [30]. From the slope of the standard curve, the amplification efficiencies for each of the four primer/probe systems were calculated using the following equation [31]:(1)$$E\left(\%\right)=\left[10\left(\frac{-1}{slope}\right)-1\right]\times 100\%$$

The percentage of pork, chicken, beef, and sheep was calculated according to the following equation: (2)$$C\left(DNA\;spec\right)=10^{\frac{Ctspec-b}{slope}}$$
where C_DNA spec_ is the concentration of the specific (pork, chicken, beef, and sheep) DNA; Ct_spec_ is the Ct value determined using the duplex assay; and b is the intercept of the standard curve of the duplex real-time PCR assay.

### 2.6. Sensitivity, Accuracy, and Precision

Sensitivity, precision, and accuracy were evaluated to determine the performance of the duplex qPCR method in quantifying chicken, pork, beef, and sheep fractions in meat products. The accuracy of the assay is determined by comparing the mean estimated value obtained from assay results to the true value and is expressed as bias or error (%), which is calculated as follows: (mean estimated value − true value)/true value × 100 [32]. Precision is the relative standard deviation (RSD) of the assay results under repeatability conditions and is calculated as RSD = standard deviation/mean value. Precision refers to the agreement among assay results. The acceptance criteria for both accuracy and precision were set within the 25% range according to guidelines from ISO documents [31,33].

### 2.7. Limit of Detection (LOD) and Limit of Quantification (LOQ)

The LOD is typically defined as the lowest analyte concentration at which ≥95% of the positive replicates are detected, thus ensuring less than 5% false negative results; the LOQ is defined as the lowest amount of the analyte in a sample that can be reliably quantified within an acceptable level of precision and accuracy [32].

### 2.8. Quantification with Calibration Factors

To establish the calibration factors for each meat species, DNA was extracted from each meat species and reference materials were adjusted to a DNA concentration of 20 ng/µL using elution buffer. The DNA from each meat species were diluted from 100% according to Table 3 and used as standard materials to obtain standard curves. Herring sperm DNA solution (20 ng/µL) (Thermo Fisher Scientific, Waltham, MA, USA) was used as a diluent. The DNA from the reference materials RefA–RefG were used to derive the respective calibration factors for each meat species as previously described [19,24]. The respective calibration factors were then used to calculate the meat content in the samples.

DNA content calculations were performed either using the Design and Analysis 2 Software (QuantStudio™ 6 Pro Real-Time PCR System, Thermo Fisher Scientific, Waltham, MA, USA), or Excel (Microsoft Office 2019, Redmond, WA, USA).

### 2.9. Meat Species Identification and Quantification Using Commercial Kits

RapidFinder™ Meat ID kits (Thermo Fisher Scientific Inc., Waltham, MA, USA) for pork, chicken, beef, and sheep were used to verify the authenticity of the reference meat samples according to the manufacturer’s instructions. A RapidFinder™ Meat Quant Multi-Meat Set (Thermo Fisher Scientific Inc., Waltham, MA, USA) was used with the RapidFinder™ Meat ID kits for pork, chicken, beef, and sheep to determine the percentage of a target meat species with respect to total meat in food and feed samples. The percentage of pork/chicken/beef/sheep DNA can be calculated using the following formula:% Species DNA = Species DNA × 100/Animal DNA.

## 3. Results and Discussion

### 3.1. Design of the Duplex Real-Time PCR System

Accurate meat quantification is crucial in standard food testing methods for regulatory purposes. While several multiplex qPCR systems have been developed for targeting four or more meat species, their accuracy is relatively low [19,20,23,24,25]. For instance, a five-plex meat speciation assay can have accuracy levels even up to 167% [24]. The inadequate accuracy may be attributed to variations in amplification efficiencies among different target sequences and reagent competition within the reaction, which can introduce biases when comparing meat species targets [24]. To address these challenges, the duplex qPCR approach offers a solution by including two specific targets with similar amplification efficiencies in the same reaction, thereby minimizing biased results. Based on the singleplex qPCR performance results for pork, chicken, beef, and sheep (Appendix A), the duplex qPCR assays were designed to precisely quantify the content of pork and chicken as one pair, and beef and sheep as another pair, in meat products.

The design of the assay target short-length amplicons selected from single-copy nuclear DNA genes for each meat species effectively addressed two challenges in meat quantification. Firstly, the performance of quantification assay using mitochondrial DNA as targets in qPCR may be hindered by the diversity of mitochondrial DNA copy numbers between different animal species and tissue types [24,34]. Secondly, the potential degradation of DNA during food processing can adversely impact the amplification of longer amplicons. It has been reported that small amplicons with fragments below 150 bp are recommended due to their increased sensitivity (LOD) in qPCR assays compared to longer fragments exceeding 200 bp [34,35].

### 3.2. Specificity of the Duplex Primer/Probe Systems

Specificity is a prerequisite for all applied PCR systems. To evaluate the specificity of the two duplex primer/probe systems, DNA isolated from a wide range of animal species and other common food ingredients were used. Animal species include cattle (*Bos taurus*), chicken (*Gallus gallus*), crocodile (*Crocodylus niloticus*), donkey (*Equus asinus*), duck (*Anatidae*), goat (*Capra hircus*), goose (*Anserinae*), hare (*Lepus europaeus*), horse (*Equus caballus*), kangaroo (*Macropodidae*), ostrich (*Struthio camelus*), pig (*Sus scrofa domestica*), rabbit (*Oryctolagus cuniculus*), red deer (*Cervus elaphus*), roe deer (*Capreolus capreolus*), sheep (*Ovis aries*), turkey (*Meleagris gallopavo*), and wild boar (*Sus scrofa scrofa*). Other common food ingredients: black mustard (*Brassica nigra*), broccoli (*Brassica oleracea*), carrot (*Daucus carota*), celery (*Apium graveolens*), chili pepper (*Capsicum* sp.), garlic (*Allium sativum*), ginger (*Zingiber officinale*), kailan (*Brassica oleracea var. alboglabra*), onion (*Allium cepa*), parsley (*Petroselinum crispum*), rapeseed (*Brassica napus*), rice (*Oryza sativa*), rosemary (*Rosmarinus officinalis*), rye (*Secale cereale*), tomato (*Solanum lycopersicum*), wheat (*Triticum aestivum*), cabbage (*Brassica oleracea var. oleracea*), and shiitake mushroom (*Lentinula edodes*). The eukaryotic system, 18S rRNA gene, was used as internal positive amplification control (PC) to determine the amplifiability of DNA extracts, and positive amplification by eukaryotic system was obtained for all species (Appendix A).

Our results showed that the chicken-, beef-, and sheep-specific primers and probes exhibited 100% specificity with no cross amplification of DNA from other species (Appendix A). However, we observed crossreactivity in the pork assay when using wild boar as a template since the target region is a conserved region for both pig and wild boar. Previous studies have also reported such crossreactivity among closely related animal species [36,37]. This observation may not be significant in this study as the focus is on the quantification of meat fractions, and wild boar contamination is less likely to occur as it is a rare and more expensive meat source. This crossreactivity will need to be considered when determining meat products containing meat from wild animals.

### 3.3. Sensitivity, Precision, and Accuracy

To assess the sensitivity of our method, LOD and LOQ of the duplex real-time PCR assays were determined by analyzing DNA mixtures in 24 replicates following the recommendations of the European Network of GMO (Genetically Modified Organisms) Laboratories (ENGL) [32]. For the determination of the LOD, we generated standard curves for each meat-specific system using binary DNA model mixtures containing known amounts of the respective meat species DNA (Table 3). The duplex standard was constructed by plotting the mean Ct values for eight dilutions (as outlined in Table 3), representing samples containing both chicken and pork, as well as beef and sheep (Figure 1). Each data point was analyzed six times (N = 24) in four different runs conducted on separate days to ensure repeatability and reproducibility. The LOD was defined as the lowest DNA concentration that led to Ct values < the Ct values obtained for crossreacting species in 23 out of 24 replicates. The LOQ was defined as the lowest concentration which could be determined with an RSD ≤ 25%.

Our data revealed that both LOD and LOQ for raw meat samples of all target species were 0.1% (0.1 ng) DNA, as depicted in Table 4 and Figure 1 and Figure 2. While for boiled meat samples, we observed a delay in DNA amplification compared to raw meat samples, as shown in Figure 1. This delay can be attributed to the autoclave treatment used during the preparation of boiled meat, resulting in increased cycle threshold (Ct) values and intercepts of the calibration curves. As a result, the LOD and LOQ for boiled meat samples of all target species was determined to be 0.32% (0.32 ng) DNA (Appendix A). It is worth noting that negative controls containing only herring sperm DNA (20 ng/µL) did not produce positive results, confirming the specificity of our method. 

To assess the performance of the method, the amplification efficiency (AE) and correlation coefficient (R^2^) were compared across the four independent runs. As indicated in Table 4 and Figure 2, our method exhibited high PCR amplification efficiency values ranging from 98.7% to 104.5%, R^2^ values ranging from 0.9873 to 0.9999, and slope values of calibration ranging from −3.279 to −3.405. These values comply with the guidelines established by the ENGL, which state that the R^2^ should exceed 0.98, the PCR efficiency should be within the range of 90% to 110%, and the slope should be approximately between ~3.1 and ~3.6 [32].

We further compared the aforementioned parameters of our method with two similar studies [24,25], it was observed that our method demonstrates superior PCR amplification efficiency, as well as a stronger correlation with higher precision and accuracy (Appendix A). This can be attributed to the utilization of duplex qPCR, which reduces variations in amplification efficiencies and minimizes biased results, as mentioned earlier. These results provide compelling evidence that the outstanding performance of our method in detecting all four meat species in both raw and processed meat samples, demonstrating its alignment with the stringent criteria established by ENGL for reliable meat analysis.

### 3.4. Determination of Calibration Factors with Reference Materials

To determine the meat content from the qPCR-measured DNA content, four independent rounds of analysis were performed using mixed reference materials DNA with known proportions of four meat species (Table 1). The calibration factors were derived by employing duplex qPCR and DNA dilutions of raw and boiled meat according to Table 3. For each analysis round, the calibration factors were determined using the meat proportion calculation formular that yielded the most accurate results. This process was repeated iteratively until the difference between the true value and the calculated value reached a minimum level [19,24,25] (Table 5). The meat proportion calculation formula (example for pork):$$pork\;content\left(\%\right)=100*\frac{Mp}{CFp}/(\frac{Mp}{CFp}+\frac{Mc}{CFc}+\frac{Mb}{CFb}+\frac{Ms}{CFs})$$
where Mp is the measured pork DNA content, CFp is the calibration factor for pork, p is pork, c is chicken, b is beef, and s is sheep.

The calculated calibration factors are shown in Table 6. It is noteworthy that, despite the observed delayed amplification in boiled meat compared to raw meat, the calibration factors exhibited relatively similar values for both types. This similarity can be attributed to the fact that the standard curve DNA sources and reference materials used for deriving calibration factors were produced under the same production conditions, resulting in comparable amplification delays. These finding indicate a consistent correlation between DNA content and actual meat proportions for both raw and processed meat.

The utilization of calibration factors distinguishes our method from numerous other reported qPCR methods that rely on mixed meat references as standards for each analysis, leading to the need for preparing reference materials for each run [20,28,29,34,35]. In contrast, our approach utilizes calibration factors derived from reference materials during method validation, thereby eliminating the need for preparing reference materials for each analysis. By employing these calibration factors, we directly convert the qPCR-measured DNA content of an unknown sample into meat weight [19,24,25]. This not only greatly enhances efficiency but also saves time in the process. 

As indicated in Table 5, most of the measured values of the major components (15% to 75%) of the raw and boiled reference materials showed good correlation with the true values. Although the precision and accuracy of the minor components (1% and 9%) were lower compared to the major components (Table 5), they were still within the acceptable range [31,33,38]. The inferior precision and accuracy of the minor components could be attributed to the inhomogeneity of the sample as it is challenging to homogenize the connective tissue and fatty meat in the sample, especially for the minor components. Similar observations have been reported in other studies [22,23,24], indicating that sample inhomogeneity in minor components can affect precision and accuracy.

We conducted a comparative analysis between our method and two other approaches that similarly utilized calibration factors [24,25], summarizing precision and accuracy ranges for each method (Appendix A). The findings demonstrated that our duplex qPCR method outperforms other reported qPCR meat quantification methods by exhibiting significantly higher precision and accuracy. This superiority holds significant importance as it aligns with the strict requirements established by regulatory authorities responsible for overseeing meat safety control. In the two multiplex qPCR studies, the amplification efficiency of different targets was compromised due to variations in their individual amplification efficiencies within the multiplex setup, leading to biased results [24,25]. However, it is important to note that the primary objective of those studies was for early screening and initial quantification of meat adulteration, making multiplex qPCR suitable for the simultaneous and rapid detection of four–five different meat species, albeit with slightly lower accuracy. In contrast, our study aimed to provide data for meat import control and enforcement actions against adulteration. Therefore, ensuring the accuracy of the data was our priority. This further underscores that different methods have distinct focuses depending on their specific objectives, and our newly developed method aligns perfectly with our intended purpose.

### 3.5. Analysis of Proficiency Test Samples and Market Monitoring Samples 

To further verify the efficacy of our new method, we analyzed three PT samples (Appendix A) using the newly developed duplex qPCR systems with the calibration factors. As all PT samples were raw materials, DNA dilutions of raw meat were used as a calibrator. The meat contents of the three samples were measured and compared to their declared meat compositions to determine the precision and accuracy of the method (Table 7).

Similar to the reference materials, the measured values of the major components of the three PT samples were highly correlated with their true values, with higher precision and accuracy. However, for the minor components (3–5%), the accuracy is lower than that of the major component. This observation is consistent with findings from other reported studies [22,23,24]. The results were further compared against those obtained using the commercial kits from Thermo Fisher Scientific RapidFinder™ Meat ID Kits and the RapidFinder™ Meat Quant Multi-Meat Set, which notably expressed results as DNA content rather than meat content. Table 7 illustrates that the duplex qPCR outcomes exhibited superior precision and accuracy when contrasted with the outcomes produced by the RapidFinder™ kit. It is important to highlight that our new method demonstrated substantially higher accuracy, particularly for minor components (3–5%), where the RapidFinder™ kit’s accuracy exceeded 25%, considerably lower than our approach. Our results suggest that the newly developed duplex qPCR method is highly effective and reliable for quantifying meat content in raw samples. Nevertheless, it is essential to acknowledge that this method does not cover all meat species present in this PT sample, such as goat, turkey, and equine, which are less commonly consumed in Singapore. Therefore, there is a need to develop accurate quantification strategies for these uncommon meat species in order to better support safe food supply in Singapore.

The method was also applied for meat fraud market monitoring testing and ensuring compliance with meat permits for meat import. The results of meat speciation and quantification results for selected street food samples were presented in Table 8. Our market monitoring testing revealed instances of food outlets engaging in fraudulent practices by substituting high-priced mutton for low-priced beef for economic gains (sample 1, 8, and 10). Additionally, during the meat permit compliance check, we identified two products, both of which were instant noodles, containing more than 5% meat (sample 11 and 12). The vendors were promptly informed and requested to obtain the necessary permits before selling these products in Singapore. These results highlight the practical applicability of our method in real-life scenarios, effectively combating meat fraud and enhancing meat safety control measures.

### 3.6. Future Perspectives

Moving forward, we identify several areas of future work to better assist meat safety control and combat meat fraud. One crucial aspect is the development of methods capable of detecting a wide variety of meat species, including uncommon species, such as goat, turkey, and donkey, that may be used as substitutes for economic gains, which is essential for comprehensive fraud detection. Furthermore, we will explore new techniques like dPCR, which enables absolute measurement of nucleic acid concentration without relying on standard curves with high sensitivity and accuracy, thereby enhancing meat quantification. By addressing these aspects and continuously improving our detection methods, we can strengthen the overall capability to detect and prevent meat fraud, ultimately contributing to enhanced food safety measures. 

## 4. Conclusions

We have successfully developed an innovative and highly accurate meat quantitative method in raw and processed meat products using duplex qPCR assays. This method utilizes calibration factors derived from raw and boiled meat references during method validation, eliminating the need for mixed meat references for each analysis and saving time. Compared to several other studies that also employed calibration factors, our method exhibits exceptional accuracy and meets regulatory requirements for meat safety control. It can be relied upon for precise and reliable meat content quantification. Moreover, our study pioneers the use of this method in monitoring meat product imports in Singapore, helping prevent illegal importation by businesses without the necessary permits. 

In conclusion, our duplex qPCR method holds great potential for regulatory testing, where accurate meat quantification is crucial to ensure compliance with regulations pertaining to meat contents. By implementing this method in laboratories, the accuracy and reliability of meat quantification can be significantly enhanced, contributing to the overall safety and authenticity of meat products in the market. 

## Figures and Tables

**Figure 1 foods-12-02971-f001:**
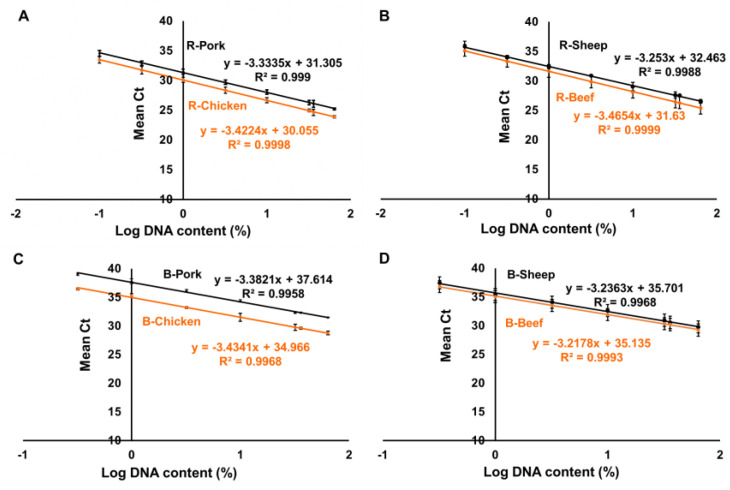
Standard curves plotted using the mean Ct values against eight dilution series (Table 3) for raw meat using the duplex PCR systems. (**A**) Standard curves for raw pork and chicken; (**B**) Standard curves for raw beef and sheep. (**C**) Standard curves for boiled pork and chicken; (**D**) Standard curves for boiled beef and sheep. R: Raw meat; B: Boiled meat; x: the log DNA amount for each meat species; y: mean Ct values; R^2^: correlation coefficient.

**Figure 2 foods-12-02971-f002:**
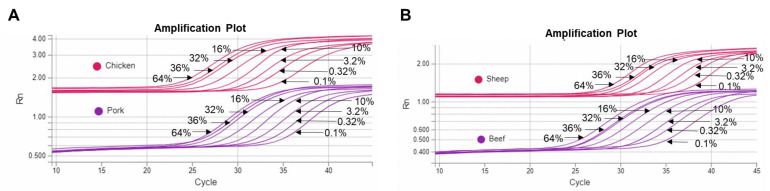
The real-time PCR data of amplification curves for raw meat using serial diluted DNA. (**A**) Amplification curves for raw chicken and pork; (**B**) Amplification curves for raw sheep and beef. For serially diluted DNA, 0.1 ng of DNA was detected for all raw meat species.

**Table 1 foods-12-02971-t001:** Composition of reference materials ranging from 1% to 75% proportion for each species in total meat ingredients.

Reference Materials Composition	Pork (%)	Beef (%)	Chicken (%)	Sheep (%)
RefA	1	9	15	75
RefB	9	1	75	15
RefC	15	75	1	9
RefD	25	25	25	25
RefE	50	0	50	0
RefF	0	50	0	50
RefG	75	15	9	1

**Table 2 foods-12-02971-t002:** Primer and probe sequences used for duplex qPCR systems.

Species	Primer/Probe	Final Conc. (nM)	Sequence	Amplicon	Gene
Pork	Sus1 F	200	CGAGAGGCTGCCGTAAAGG	80	Beta-actin-gene DQ452569 [23]
	Sus1 R	200	TGCAAGGAACACGGCTAAGTG		
	Sus1-VIC	100	VIC-TCTGACGTGACTCCCCGACCTGG-BHQ1		
Chicken	Gallus1 F	200	CAGCTGGCCTGCCGG	76	TF-GB3 X6009 [23]
	Gallus1 R	200	CCCAGTGGAATGTGGTATTCA		
	Gallus1-FAM	100	FAM-TCTGCCACTCCTCTGCACCCAGT-BHQ1		
Sheep	OA-PRLR-F	200	CCAACATGCCTTTAAACCCTCAA	88	Prolactin receptor [19]
	OA-PRLR-R	200	GGAACTGTAGCCTTCTGACTCG		
	OA-PRLR FAM	100	FAM-TGCCTTTCCTTCCCCGCCAGTCTC-BHQ1		
Beef	Rd 1 F	200	GTAGGTGCACAGTACGTTCTGAAG	96	Beta-actin-gene EH170825 [19]
	Rd 1 R	200	GGCCAGACTGGGCACATG		
	Rd 1-VIC	100	VIC-GAACCTCATTCTGGGGCCCCG-BHQ1		
18S	18S-F	200	CTGCCCTATCAACTTTCGATGGTA	113	18S rRNA [28,29]
	18S-R	200	TTGGATGTGGTAGCCGTTTCTCA		
	18S-FAM	100	FAM-ACGGGTAACGGGGAATCAGGGTTCGATT-BHQ1		

FAM: 6-carboxyfluorescein, VIC: 2′-chloro-7′-phenyl-1, 4-dichloro-6-carboxyfluorescein, BHQ 1: Black Hole Quencher 1.

**Table 3 foods-12-02971-t003:** Dilution of Standards for Duplex Chicken and Pork qPCR and Duplex Sheep and Beef qPCR.

STD	Duplex Chicken and Pork qPCR		Duplex Sheep and Beef qPCR	
	Chicken %	Pork %	Beef %	Sheep %
1	64%	36%	36%	64%
2	32%	32%	32%	32%
3	10%	10%	10%	10%
4	3.2%	3.2%	3.2%	3.2%
5	1%	1%	1%	1%
6	0.32%	0.32%	0.32%	0.32%
7	0.1%	0.1%	0.1%	0.1%
8	36%	64%	64%	36%

The DNA was isolated from pure frozen meat of the appropriate species; Herring sperm DNA solution (20 ng/µL) was used as a diluent for 32–0.1%.

**Table 4 foods-12-02971-t004:** Performance of the duplex PCR.

Duplex PCR	Chicken	Pork	Sheep	Beef
Amplification efficiencies %	99.11	98.70	104.54	103.70
Correlation R^2^	0.9998	0.9945	0.9969	0.9985
RSD (%)	5.23	8.54	8.75	11.4
Bias (%)	2.98	4.12	3.4	3.1

The amplification efficiencies, correlation, precision, and accuracy were compiled from 4 independent measurements for both raw and boiled meat. The amplification efficiencies were calculated using the equation in Section 2.5. Precision is the relative standard deviation, RSD = the standard deviation/the mean value; it was calculated by averaging the individual RSD of all dilutions (see Table 3). The bias was calculated by averaging the absolute values of the individual dilutions ((mean measured value − true value)/true value × 100). The numbers shown here represent mean values from 4 independent experiments (N = 24).

**Table 5 foods-12-02971-t005:** Values of the raw and boiled reference materials measured by duplex qPCR with calibration factors.

Meat Compositions (%)		1	9	15	25	50	75
Raw Pork	Md (%)	1.24	9.89	16.32	25.04	48.85	72.02
	RSD (%)	20.42	7.49	7.62	13.24	7.6	14.01
	Bias (%)	24.00	9.89	8.80	0.16	−2.30	−3.97
Boiled Pork	Md (%)	0.91	9.93	16.52	23.68	45.89	70.76
	RSD (%)	24.58	22.48	10.85	17.16	14.47	8.03
	Bias (%)	−9.00	10.33	10.13	−5.28	−8.22	−5.65
Raw Chicken	Md (%)	1.23	9.74	12.18	27.59	51.15	81.25
	RSD (%)	5.09	18.31	6.57	13.84	5.56	6.15
	Bias (%)	23.00	8.22	−18.80	10.36	2.30	8.33
Boiled Chicken	Md (%)	1.21	11.09	14.90	27.89	54.10	78.47
	RSD (%)	22.41	9.41	7.14	4.38	2.45	7.12
	Bias (%)	21.00	23.22	−0.67	11.56	8.20	4.63
Raw Beef	Md (%)	1.06	10.25	17.33	28.35	51.31	72.22
	RSD (%)	15.72	8.92	11.79	8.60	11.67	8.81
	Bias (%)	6.00	13.89	15.53	13.40	2.62	−3.71
Boiled Beef	Md (%)	1.15	9.89	17.33	29.20	51.45	72.47
	RSD (%)	15.72	8.92	11.79	8.60	11.67	8.81
	Bias (%)	15.00	9.89	15.53	16.80	2.90	−3.37
Raw Sheep	Md (%)	0.91	9.68	15.03	25.28	49.69	75.74
	RSD (%)	22.42	8.60	13.08	16.88	19.48	7.68
	Bias (%)	−9.00	7.56	0.20	1.12	−0.62	0.99
Boiled Sheep	Md (%)	0.96	9.71	16.60	25.86	48.51	73.64
	RSD (%)	23.02	8.60	11.79	16.88	24.98	10.16
	Bias (%)	−4.00	7.89	10.67	3.44	−2.98	−1.81

Md as mean measured value; RSD as Precision, RSD = the standard deviation/the mean value; and Bias as accuracy, bias = (mean measured value − true value)/true value × 100. The values represent four measurements on four different days.

**Table 6 foods-12-02971-t006:** Calibration factors generated with the meat composition compiled in Table 1.

	Raw				Boiled			
	Chicken	Pork	Sheep	Beef	Chicken	Pork	Sheep	Beef
1	0.87	1.04	0.84	0.97	0.91	0.99	0.88	0.88
2	0.84	0.97	0.79	0.82	0.82	0.91	0.86	0.92
3	0.89	0.86	0.98	0.95	0.86	0.97	0.94	0.85
4	0.79	0.99	0.89	0.89	0.87	0.98	0.89	0.99
Mean	0.85	0.97	0.88	0.91	0.87	0.96	0.89	0.91
RSD	5.12	7.83	9.21	7.42	4.27	3.73	3.81	6.65

Four independent rounds of analysis were performed using sample DNA with the meat composition compiled in Table 1 to derive the calibration factors. Mean calibration factors and the RSD were calculated.

**Table 7 foods-12-02971-t007:** Values of the PT samples measured using duplex qPCR systems.

Sample Name	Target	True Values (%)	Duplex PCR Results (Meat %)			RapidFinder™ Kit Results (DNA %)		
			Md (%)	RSD (%)	Bias (%)	Md (%)	RSD (%)	Bias (%)
FAPAS-PT1	Pork	3	3.70	12.64	23.33	4.26	16.1	42
	Chicken	0	0	-	-	0	-	-
	Beef	0	0	-	-	0	-	-
	Sheep	94	90.32	5.39	−3.91	87.24	10.02	−7.19
FAPAS-PT2	Pork	0	0	-	-	0	-	-
	Chicken	0	0	-	-	0	-	-
	Beef	90	89.39	17.46	−0.68	93.7	15.1	4.11
	Sheep	5	3.81	13.84	−23.8	7.02	14.6	40.40
FAPAS-PT3	Pork	3	2.41	15.27	−24.48	4.6	7.5	53.33
	Chicken	92	88.79	4.56	−3.49	87.7	6.58	−4.67
	Beef	0	0	-	-	0	-	-
	Sheep	0	0	-	-	0	-	-

Md as mean measured value, RSD as Precision, Bias as accuracy.

**Table 8 foods-12-02971-t008:** Market samples analyzed by duplex qPCR.

Study Type	S/N	Sample Name	Duplex PCR Results (Meat %)			
			Beef	Sheep	Chicken	Pork
Meat fraud market monitoring	1	RTE Mutton Briyani	71.4	28.6	0	0
	2	Raw Minced Mutton	0	99.8	0	0
	3	RTE Mutton Briyani	0	95.3	0	0
	4	RTE Mutton Curry (Boneless)	0	98.2	0	0
	5	RTE Keema (minced mutton)	0	94.9	0	0
	6	Raw Mutton (with bones)	0	98.5	0	0
	7	Raw Mutton Minced Meat (Fine)	0	99.5	0	0
	8	Raw Mutton Cube for Mutton Briyani Curry	72.7	27.3	0	0
	9	Raw Mutton Bone Cut for Mutton Curry (prata)	0	97.5	0	0
	10	RTE Mutton Curry	63.9	36.1	0	0
Meat permit compliance check	11	Chilli Pork Flavor Instant Bowl Noodles	0	0	0	10.8
	12	Chilli Pork Flavor Instant Noodles	0	0	0	10.9

## Data Availability

The data presented in this study are available on request from the corresponding author.

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
