# Peer review of "Quantification of Pork, Chicken, Beef, and Sheep Contents in Meat Products Using Duplex Real-Time PCR"

_foods, 2023, doi:10.3390/foods12152971_

Round 1
Reviewer 1 Report
Comments and Suggestions for Authors
Manuscript ID: foods-2465587
Title: Quantification of Pork, Chicken, Beef and Sheep Contents in Meat Products Using Duplex Real-Time PCR
This manuscript developed an accurate meat quantitative method by utilizing two duplex qPCR assays in combination of calibration factors derived from raw and boiled matrix adapted meat references. This manuscript is novel.
Major concern:
In “3. Results and Discussion”, the authors should explain why Duplex Real-Time PCR is better than other multiplex qPCR systems.
In “3. Results and Discussion”, the authors should deduce the accuracy of Duplex Real-Time when it applied to other species.
Author Response
Dear Reviewer,
We express our gratitude for your dedicated review of our work. Your feedback has been highly valuable in enhancing the quality of our work. The suggestions and comments provided by you have played a significant role in driving improvements. We have thoroughly considered your comments and incorporated them accordingly, along with providing detailed explanations. For specific details, please refer to the revised version of the article and our response to the review report.
Once again, we want to emphasize our genuine appreciation for recognizing the strengths of our work and offering constructive feedback. If you have any additional suggestions or questions, please don't hesitate to reach out to us. We wish you continued success in all your endeavors!
Sincerely,
Wang Yanwen, Dr.
Singapore Food Agency
7 International Business Park, Singapore 609919
Reviewer one:
Manuscript ID: foods-2465587
Title: Quantification of Pork, Chicken, Beef and Sheep Contents in Meat Products Using Duplex Real-Time PCR
This manuscript developed an accurate meat quantitative method by utilizing two duplex qPCR assays in combination of calibration factors derived from raw and boiled matrix adapted meat references. This manuscript is novel.
Major concern:
In “3. Results and Discussion”, the authors should explain why Duplex Real-Time PCR is better than other multiplex qPCR systems.
Author’s reply: Thanks for the suggestion. We added in explanations in Results and Discussion 3.1, please refer to the revised version (line 211 to 221).
In “3. Results and Discussion”, the authors should deduce the accuracy of Duplex Real-Time when it applied to other species.
Author’s reply: Thanks for the comment. Although we can deduce the content and accuracy of other species in the PT samples based on our measurements, this data does not accurately represent the performance of our duplex qPCR method for other species. Additionally, we acknowledge that developing a detection method for other species is part of our future research agenda (Line 408-412).
Reviewer 2 Report
Comments and Suggestions for Authors
The manuscript entitled “Quantification of Pork, Chicken, Beef and Sheep Contents in Meat Products Using Duplex Real-Time PCR” is a good work and it provides a promising method for the detection of meat products. However, the following issues are to be addressed,
Ø The authors should have compared the developed assay with the conventional endpoint PCR assay. A comparison should be made with sensitivity/ limit of detection. Fold differences should be mentioned in the abstract.
Ø The rationale or advantages of selecting the target genes for all 4 animal species should be mentioned in the discussion.
Ø Line 108-109: Whether the species of the animals were identified morphologically before collection?
Ø Why in meat admixture studies, the ratios were only up to 1%. For Example, the samples of 0.5%, 0.1%, 0.01%, and 0.01% pork meat spiked in beef matrix seem to be more suitable for sensitivity determination.
Ø To determine repeatability and reproducibility, intra-day (three times a day) and inter-day (three consecutive) evaluations should be done.
Ø The introduction is poorly written. There is too much unnecessary and confusing content that is not related to the topic of the article. For example, Calibration factors, matrix-specific multiplication factors, matrix-adapted reference material, etc. What do you mean? Rewrite.
Ø Introduction: The description of the importance of developing TaqMan real-time PCR assay for meat authentication is not clear and concrete enough. A better organization of the description is suggested.
Ø The discussion of "Results and discussion" is lacking. In this stage, it seems to be more of a description of the results.
Ø This method should be compared with the reported nucleic acid amplification assays used for meat authentication, which should also be summarized in the table.
Ø Did the authors tried to estimate minimal adulteration values? The given now 1% is not a record value. LOD (………. fg or pg of DNA) is required for all species.
Ø It will be useful to give a comparative table at the end of the discussion for better justification of the reached improvements via comparison of quantitative parameters for the proposed and other techniques described as tools to control adulteration of meat products.
Ø Are there any limitations to TaqMan technology? Or what aspects of this technology need to be improved? Especially for meat authentication. The author is expected to discuss future prospects in the conclusion.
Ø Language is poor and the manuscript should not be printed in this form. The writing should be polished for accuracy and clarity.
Comments on the Quality of English LanguageLanguage is poor and the manuscript should not be printed in this form. The writing should be polished for accuracy and clarity
Author Response
Response to Reviewer 2 Comments
Dear Reviewer,
We express our gratitude for your dedicated review of our work. Your feedback has been highly valuable in enhancing the quality of our work. The suggestions and comments provided by you have played a significant role in driving improvements. We have thoroughly considered your comments and incorporated them accordingly, along with providing detailed explanations. For specific details, please refer to the revised version of the article and our response to the review report.
Once again, we want to emphasize our genuine appreciation for recognizing the strengths of our work and offering constructive feedback. If you have any additional suggestions or questions, please don't hesitate to reach out to us. We wish you continued success in all your endeavors!
Sincerely,
Wang Yanwen, Dr.
Singapore Food Agency
7 International Business Park, Singapore 609919
Reviewer two:
The manuscript entitled “Quantification of Pork, Chicken, Beef and Sheep Contents in Meat Products Using Duplex Real-Time PCR” is a good work and it provides a promising method for the detection of meat products. However, the following issues are to be addressed,
Ø The authors should have compared the developed assay with the conventional endpoint PCR assay. A comparison should be made with sensitivity/ limit of detection. Fold differences should be mentioned in the abstract.
Author’s reply: Thanks for the comment. The main objective of this method is to accurately quantify the meat content in a sample, thereby supporting meat safety control and preventing food fraud. By utilizing the rapid duplex qPCR method, it significantly reduces the need for manual efforts compared to conventional endpoint PCR. With the amplification of specific DNA sequences associated with each meat species, we are able to achieve precise determination of meat quantity. In contrast, conventional endpoint PCR requires additional post-amplification analyses such as gel electrophoresis or hybridization to confirm the presence or absence of target amplicons, making accurate meat content quantification impractical. Our study has demonstrated the accuracy and sensitivity of the newly developed method for meat quantification, aligning perfectly with our intended purpose. Therefore, in this particular investigation, there is no need for a comparison between the real-time PCR assay and conventional endpoint PCR.
Ø The rationale or advantages of selecting the target genes for all 4 animal species should be mentioned in the discussion.
Author’s reply: Thanks for the comment. The rationale for selecting the four animal species is mentioned in the introduction largely to cater the needs of this assay development (Lines 98-100), hence, this information was not in the discussion section.
Ø Line 108-109: Whether the species of the animals were identified morphologically before collection?
Author’s reply: Thank you for your question. In Singapore, meat products are sourced from different countries and undergo initial processing to facilitate import, hence it is not possible to differentiate the meat types based on morphological analysis before collection. To ensure the authenticity of the meat, validated commercial test kits were used to confirm the samples prior the assay development. The details can be found in material methods (Line 107 to 110).
Ø Why in meat admixture studies, the ratios were only up to 1%. For Example, the samples of 0.5%, 0.1%, 0.01%, and 0.01% pork meat spiked in beef matrix seem to be more suitable for sensitivity determination.
Author’s reply: Thanks for the comment. We agree with your point that conducting sensitivity testing using spiked meat samples is crucial. In the development of this assay, our goal is to accurately quantify the meat content to support meat safety control measures (where food products with more than 5% meat content require a valid import license and permit) and prevent meat fraud. Therefore, we intentionally spiked the samples up to a ratio of 1% to align with our intended use. Our focus is not on establishing an assay with extremely low detection limits, such as 0.1% or 0.01%, as such low levels could potentially be attributed to cross-contamination during food production lines.
Ø To determine repeatability and reproducibility, intra-day (three times a day) and inter-day (three consecutive) evaluations should be done.
Author’s reply: Thanks for the comment. The repeatability and reproducibility of this study were determined following the guidelines of IS0 20813:2019: Molecular biomarker analysis — Methods of analysis for the detection and identification of animal species in foods and food products (nucleic acid-based methods) — General requirements and definitions. Specifically, each data point was analyzed six times (N = 6) in four different runs conducted on separate days by two analysts to ensure multiple measurements under controlled conditions to assess repeatability and reproducibility. Additionally, the method was validated using proficiency test samples and market samples, and the performance was better than commercial method currently adopted in the lab (Table 7). Considering these factors, we conclude that the repeatability and reproducibility of the study's findings are reliable and meet the necessary requirements.
Ø The introduction is poorly written. There is too much unnecessary and confusing content that is not related to the topic of the article. For example, Calibration factors, matrix-specific multiplication factors, matrix-adapted reference material, etc. What do you mean? Rewrite.
Author’s reply: Thanks for the comments. We have amended the introduction for better clarity, please refer to the revised version for further details (Line 81-95).
Ø Introduction: The description of the importance of developing TaqMan real-time PCR assay for meat authentication is not clear and concrete enough. A better organization of the description is suggested.
Author’s reply: Thanks for the suggestion. We agree that TaqMan real-time PCR assays have played a transformative role in nucleic acid analysis, including the detection and quantification of meat species. We have incorporated a short description on the importance of this method in the introduction section of our study. Please refer to the revised version for further details (line 74-78).
Ø The discussion of "Results and discussion" is lacking. In this stage, it seems to be more of a description of the results.
Author’s reply: Thank you very much for your suggestion. We have strengthened the discussion section and made the necessary modifications accordingly. Please refer to the revised version for more details.
Ø This method should be compared with the reported nucleic acid amplification assays used for meat authentication, which should also be summarized in the table.
Author’s reply: Thanks for the comment. We have added two tables in the revised version. Table S4 compares the performance of our method with two similar studies, highlighting the differences in key parameters. Table 7 presents a comparison between the proficiency testing (PT) results generated by our method and those obtained using a commercial kit. These additional comparisons provide a comprehensive assessment of our method's performance.
Ø Did the authors tried to estimate minimal adulteration values? The given now 1% is not a record value. LOD (………. fg or pg of DNA) is required for all species.
Author’s reply: Thanks for the comment. The limit of detection (LOD) for the duplex qPCR assays was determined using binary DNA model mixtures, as shown in Table 3. Our findings revealed that the LOD for raw meat samples of all target species is 0.1% (0.1 ng) DNA, as depicted in Table 4 and Figures 1 and 3. However, for boiled meat samples, beef and pork DNA at 0.1% (0.1 ng) were not detected, as shown in Figure 2 and Figure S1. Consequently, the LOD for boiled meat samples of all target species is 0.32% (0.32 ng) DNA (Line 284-290).
Ø It will be useful to give a comparative table at the end of the discussion for better justification of the reached improvements via comparison of quantitative parameters for the proposed and other techniques described as tools to control adulteration of meat products.
Author’s reply: Thank you for the suggestion. We have included a comparison table to highlight the performance of our method (Table S4) (Line 306-313 and Line 344-357). Please refer to the revised version for more details.
Ø Are there any limitations to TaqMan technology? Or what aspects of this technology need to be improved? Especially for meat authentication. The author is expected to discuss future prospects in the conclusion.
Author’s reply: Thanks for the question and suggestion. Although TaqMan technology offers several advantages in DNA analysis, it is important to consider certain limitations, especially in the context of food authentication. Firstly, the selection of appropriate targets and optimization of assay design are critical for effective meat fraud detection. Choosing conserved regions within the target species' genome and incorporating species-specific markers can enhance assay specificity and sensitivity. Optimization of assay design, including primer and probe design, as well as PCR parameters, can improve the efficiency of target amplification and detection. Secondly, it is crucial to expand the range of target species to address the evolving nature of meat fraud. Developing TaqMan assays capable of detecting a wide variety of meat species enables comprehensive fraud detection, including the identification of less common or exotic species that may be used as substitutes. Thirdly, the establishment and utilization of validated reference materials and standards play a vital role in accurate meat fraud detection. These reference materials should consist of certified DNA samples or genomic libraries representing a diverse range of meat species. They can be used for assay validation, quantification, and quality control to ensure the accuracy and comparability of results across different laboratories. We included future prospects in the conclusion part (Line 408-417).
Ø Language is poor and the manuscript should not be printed in this form. The writing should be polished for accuracy and clarity.
Author’s reply: Thanks for the comment. The formatting of the article adheres to the requirements of the journal. Additionally, we have engaged the professionals to proofread the English section of the paper to enhance the accuracy for better clarity.
Comments on the Quality of English Language
Language is poor and the manuscript should not be printed in this form. The writing should be polished for accuracy and clarity
Author’s reply: Thank you for the comment. we have engaged the professionals to proofread the English section of the paper to enhance the accuracy for better clarity.

Round 2
Reviewer 2 Report
Comments and Suggestions for Authors
None
Comments on the Quality of English LanguageMinor editing of English language required
Author Response
Dear Reviewer,
Thank you for your comments. we have revised the manuscript based on your suggestion, please refer to the revision.